# Epidemiology of Bronchiolitis and Respiratory Syncytial Virus and Analysis of Length of Stay from 2015 to 2022: Retrospective Observational Study of Hospital Discharge Records from an Italian Southern Province before and during the COVID-19 Pandemic

**DOI:** 10.3390/diseases12010017

**Published:** 2024-01-05

**Authors:** Fabrizio Cedrone, Vincenzo Montagna, Livio Del Duca, Laura Camplone, Riccardo Mazzocca, Federica Carfagnini, Angela Ancona, Omar Enzo Santangelo, Valterio Fortunato, Giuseppe Di Martino

**Affiliations:** 1Hospital Healthcare Management, Local Health Autority of Pescara, Via Renato Paolini, 65124 Pescara, Italy; livio.delduca@asl.pe.it (L.D.D.); federica.carfagnini@asl.pe.it (F.C.); valterio.fortunato@asl.pe.it (V.F.); 2Postgraduate School of Hygiene and Preventive Medicine, Università Politecnica delle Marche, 60100 Ancona, Italy; s1092987@pm.univpm.it; 3Postgraduate School of Hygiene and Preventive Medicine, University of L’Aquila, 67100 L’Aquila, Italy; laura.camplone@graduate.univaq.it (L.C.); riccardo.mazzocca@graduate.univaq.it (R.M.); 4School of Hygiene and Preventive Medicine, Vita-Salute San Raffaele University, 20132 Milan, Italy; ancona.angela@hsr.it; 5Regional Health Care and Social Agency of Lodi, ASST Lodi, 26900 Lodi, Italy; omarenzosantangelo@hotmail.it; 6Department of Medicine and Ageing Sciences, “G. d’Annunzio” University of Chieti-Pescara, 66100 Chieti, Italy; 7Unit of Hygiene, Epidemiology and Public Health, Local Health Authority of Pescara, 65100 Pescara, Italy

**Keywords:** bronchiolitis, respiratory syncytial virus, hospital discharge records, in-hospital length of stay, epidemiology, infectious disease, public health

## Abstract

Background: Severe respiratory infections, including pneumonia or bronchiolitis, caused by RSV can range from mild upper respiratory tract infections to those leading to hospitalization and serious complications such as respiratory failure in children. High-risk groups, such as premature infants and infants with underlying medical conditions, have a higher susceptibility to severe RSV disease. We conducted a retrospective study from years 2015 to 2022 in the Local Health Authority (LHA) of Pescara that counts about 320,000 inhabitants, with the aim to evaluate the burden of RSV infection, focusing on the incidence, hospitalization, and characteristics that may prolong hospital stays. Methods: All hospitalizations from 2015 to 2022 were extracted from the hospital discharge record. The monthly hospitalization rates were calculated and standardized by gender and age for the population resident in the Province of Pescara on 1 January 2015. Results: During the study period, 31,837 admissions were reported among patients aged less than 6 years. Of those, 520 hospitalizations were referred for bronchiolitis. Monthly admission rates highlighted the seasonality of bronchiolitis admissions, with higher rates in the months from December to March in all study years included. The winter seasons of years 2021 and 2022 reported a surge in bronchiolitis incidence, with a rate of 4.0/1000 (95% CI 2.964–5.146) in December 2021 and 4.0 (95% CI 2.891–5.020) in December 2022. Conclusions: Bronchiolitis represents an important cause of hospitalization among patients aged less than 6 years. The incidence was particularly increased during the winter seasons in years 2021 and 2022.

## 1. Introduction

One of the major causes of acute lower respiratory tract infection (ALRI) morbidity and postnatal mortality in children younger than 5 years of age is the infection caused by the respiratory syncytial virus (RSV), which is one of the leading causes of infant hospitalization worldwide [1]. Severe respiratory infections caused by RSV can range from mild upper respiratory tract infections to pneumonia or bronchiolitis, which can lead to hospitalization and serious complications such as respiratory failure in children. High-risk groups, such as premature infants and infants with underlying medical conditions like chronic lung disease or bronchopulmonary dysplasia, hemodynamically significant congenital heart disease, immunocompromised conditions, or severe neuromuscular disease, have a higher susceptibility to severe RSV disease, resulting in increased morbidity and mortality rates compared to those without these conditions [2,3]. Other known risk factors for hospitalization among patients with RSV infection are low birth weight, male gender, presence of an older sibling, exposure to smoking, young maternal age, and living in suburban context [4]. Due to its highly contagious nature, RSV is transmitted via contact with oral or nasopharyngeal secretions. Following an incubation period of 4–6 days, the infection typically starts with flu-like symptoms and affects the upper respiratory tract, leading to nasal congestion, rhinorrhea, and cough. Infants under the age of 2 have a higher risk of developing bronchiolitis, an inflammation of the small lung airways, accompanied by coughing and breathing difficulties. Neonates and infants may also exhibit symptoms like pneumonia and wheezing [5]. Beyond the acute phase, RSV infections in the first year of life, even without hospitalization, increase the risk of recurrent wheezing and asthma development [6]. A recent study showed that patients hospitalized for RSV infections in the first 2 years of age had a three-fold higher risk of asthma hospitalization and greater use of anti-asthmatic drugs [7]. 

In 2019, RSV infections resulted in 33 million episodes worldwide, with 3.6 million hospital admissions, 26,300 in-hospital deaths, and 101,400 RSV-related overall deaths. Over 95% of episodes and more than 97% of deaths occurred in low-income and middle-income countries across all age groups. Among ALRI, there were 6.6 million RSV-associated infection episodes in infants aged 0–6 months, with 1.4 million hospital admissions, 13,300 in-hospital deaths and 45,700 RSV-attributable overall deaths [8]. Globally, RSV is one the leading causes of death among respiratory tract infections, resulting in 160,000 deaths annually worldwide [9]. RSV infection in healthy adults has typically resulted in mild illness, but recent studies have highlighted the potential role of RSV in the genesis of severe disease in the adult population, particularly among older patients or high-risk adults with conditions such as chronic pulmonary diseases and congestive heart failure [10].

The epidemiology of RSV depends also on geographic location and climate characteristics. In Italy usually, the virus typically spreads during the period from October/November to March/April. The peak incidence occurs in January/February, partially coinciding with the flu season [11].

Regarding the preventive aspects, the only agent currently approved for young children for the prevention of RSV, palivizumab, is indicated only for children with certain conditions, for example, preterm or high-risk co-morbidities [12]. In May 2023, the U.S. Food and Drug Administration approved a vaccine against respiratory syncytial virus (RSV) for the prevention of lower respiratory tract disease caused by RSV in subjects aged 60 years and older [13].

Although the World Health Organization (WHO) has started a global effort in developing international standards for RSV surveillance, actually in the EU area, surveillance systems are fragmentary, involving only 20 out of 27 member states, with heterogeneity in the modality of collecting data [14]. Italy improved the surveillance system for influenza-like illnesses (ILI) during flu season, including also other lower respiratory tract illnesses (LRTI) such as RSV only from season 2022 [14]. This surveillance system actually integrates the epidemiological data with virological test results, aiming to evaluate the spread of respiratory tract pathogens. It can be helpful to evaluate the RSV burden of diseases but, on the other hand, is not able to give information on previous years, and it lacks in the evaluation of patients’ outcomes.

In Italy, hospital discharge records (HDRs) serve as invaluable tools for assessing the impact of various diseases on healthcare costs and utilization [15,16,17]. These records encompass demographic data, clinical features of the hospitalization, patients’ comorbidities identified by ICD-9 CM codes, and the type of hospital discharge. HDRs, while having certain limitations, can also function as indicators of healthcare usage, giving information on patients’ outcomes and clinical characteristics of the diseases. 

We conducted a retrospective study using the data from the HDRs of 7 years (from 2015 to 2022) of activity of the Local Health Authority (LHA) of Pescara, with the aim of evaluating the burden of RSV infection, focusing on the incidence, hospitalization, and characteristics that may prolong hospital stays.

## 2. Materials and Methods

A retrospective observational study of hospital discharge records (HDRs) referred to the Local Health Authority (LHA) of Pescara in the period 2015–2022 was performed. The LHA of Pescara lies within the province of Abruzzo region, serves approximately 320,000 inhabitants, and is organized in three hospitals: a tertiary referral hospital in the city of Pescara and two spokes.

The Pescara hospital is an important pediatric center with regional relevance; in fact, in addition to the provincial birth center with a neonatology department and neonatal intensive care, it has a pediatrics department that manages sub-intensive settings, and also has the only neonatal and pediatric intensive care units in the Abruzzo region.

HDRs contain various data regarding the patient’s socio-demographic information, such as sex and age and other information regarding the reason for hospitalization and the procedures performed during the hospital stay. The HDR can contain up to six diagnoses that describe the clinical characteristics of the patient (one main diagnosis and up to five comorbidities, among which concomitant morbid events are included). These diagnoses are coded according to the International Classification of Disease 9th Clinical Modification (ICD-9-CM) system, the National Center for Health Statistics (NCHS) and the Centers for Medicare and Medicaid Services External, Atlanta, GA, USA.

All HDRs from patients younger than 5 years of age and diagnosed with acute bronchiolitis in the study period were selected (ICD-9-CM: 466.1) and were attributed to respiratory syncytial virus when ICD-9-CM codes 466.11 (respiratory syncytial virus bronchiolitis) and 079.6 (respiratory syncytial virus) were present. Diagnoses were analyzed to detect the presence of some risk factors such as prematurity (ICD-9-CM: 765.20–765.28), congenital heart disease (ICD-9-CM: 745.0–747.9) and congenital defects with cardiac affectation (ICD-9-CM: 758.0–758.9), such as Patau syndrome, Down syndrome, Edwards syndrome, conditions due to autosomal chromosomal anomalies, gonadal dysgenesis, Klinefelter syndrome, and conditions due to unspecified chromosomal anomalies [18]. The transfers recorded in the HDRs were analyzed to indicate which patients needed to be transferred during hospitalization to a more intensive care setting or to a different ward.

### Statistical Analysis

Qualitative variables are expressed in terms of frequency and percentage, and quantitative variables are expressed as mean and standard deviation (SD) or median and interquartile range (IQR), according to their distribution.

The length of hospital stay (LOS) was calculated by subtracting the day of discharge from the day of hospitalization and used as the dependent variable of a negative binomial regression model to evaluate factors associated with the increase in hospital stay, and relative results are expressed as incidence rate ratios (IRRs) with relative confidence interval. *p*-values < 0.05 were considered significant.

For each of the years of observation, the monthly hospitalization rates were calculated and standardized by gender and age for the population resident in the Province of Pescara on 1 January 2015, the start date of the study observation. To calculate the monthly rate for each year, the resident population of the Province as of 1 January was considered.

Data relating to the demographic structure (sex and year) of the population were extracted from the database of the Italian National Institute of Statistics (ISTAT) website. The statistical analysis was performed with STATA v14.2 software (StataCorp LLC, College Station, TX, USA).

## 3. Results

During the study period, 31,837 admissions were reported in Pescara Hospital among patients aged less than 6 years, with 520 hospitalizations referred for bronchiolitis. Of those, the majority were male subjects (298, 57.3%), aged less than 1 year. Most of the bronchiolitis admissions required intensive care (459, 88.3%), as reported in Table 1.

Only 189 admissions (36.4%) were due to RSV. No patients died during hospitalization. Median length of stay was higher among patients with bronchiolitis, compared to all pediatric hospitalizations (4 IQR 2–9 vs. 3 IQR 2–5, *p* < 0.001), as reported in Table 2. Age and intensive care were the strongest risk factors associated with LOS (Table 3). 

Monthly admission rates highlighted the seasonality of bronchiolitis admissions, with higher rates in months from December to March in all study years included (Appendix A). Only the winter season between years 2020 and 2021 reported no cases, in parallel with the COVID-19 pandemic.

Winter seasons of years 2021 and 2022 reported a surge in bronchiolitis incidence, with a rate of 4.005/1000 (95% CI 2.964–5.146) in December 2021 and 3.956 (95% CI 2.891–5.020) in December 2022 (Figure 1).

## 4. Discussion

In Europe, estimates of RSV-related hospitalizations in children aged under 5 years, derived from national data, literature reviews, multiple imputation, and nearest neighbor matching approaches, indicate an average of 213,014 hospital admissions per winter linked to RSV in the European Union, Norway, and the United Kingdom, with a 95% confidence interval ranging from 192,181 to 233,844 [19]. In a multi-center, prospective, observational birth cohort study encompassing healthy term-born infants across four European countries (Finland, the Netherlands, Spain, and the United Kingdom—England and Scotland), the overall cohort showed an incidence of RSV-associated hospitalizations at 1.8% (95% CI: 1.6–2.1). Additionally, the incidence of RSV infection was 26.2% (95% CI: 24.0–28.6), and the incidence of medically attended RSV cases stood at 14.1% (95% CI: 12.3–16.0) [20]. The studies revealed a significant burden of RSV-related hospitalizations in children, with a particular emphasis on infants during their first year of life [21]. This observation is crucial, as it highlights the vulnerability of very young children to RSV infections, even those without pre-existing health conditions. 

The findings confirm that RSV disproportionately affects infants, premature infants, and those with underlying medical conditions. Furthermore, RSV hospitalization rates peak during the winter months, especially from December through March [22]. These risk factors and the seasonality are consistent with previous research but emphasize the need for targeted prevention strategies.

Our data revealed a notable rise in incidence during the post-COVID period. Prior to the pandemic, hospitalization rates stood at 1.5 per 1000 inhabitants. However, during the peak months of December 2021–2022 in the post-COVID era, we observed a surge to 3.7 cases per 1000 inhabitants.

In Italy, the detection rate of RSV experienced a staggering 99% decline during the 2020–2021 period, during the COVID-19 pandemic, when compared to the previous two seasons. Furthermore, the incidence of RSV among hospitalized children decreased from 38.1% to a mere 4.7% in the same 2020–2021 period [23,24]. Several other studies conducted in the aftermath of the COVID-19 pandemic and the resultant restrictions have also highlighted a significant uptick in RSV-related hospitalizations, particularly among healthy children [25]. 

The re-interruption of RSV circulation during the pandemic likely played a role in diminishing population immunity, consequently resulting in a surge of RSV cases once the restrictions were eased [26]. This scenario gives rise to the concept of ‘immune debt’ [27], wherein the disruption of RSV transmission during the pandemic has left the population with diminished immunity against RSV, leading to an increase in cases. This phenomenon underscores the critical importance of maintaining immunity through vaccination and other preventive measures. In addition, in Italy, restrictions were loosened, with the re-opening of schools and activities, facilitating the circulation of respiratory viruses.

It is known that the risk of RSV infection is usually very low during first months of life, due to maternal antibodies, but around half of children are infected within their first year of life, and the chance of having developed an RSV infection reaches almost 100% by 2 years of age [14]. The lockdown that occurred during the COVID-19 pandemic could have led to an increase in RSV infections among patients older than 2 years of age that were not involved in the infection during the 2020 and 2021 seasons.

Recently, in Europe, the European Medicines Agency (EMA) granted approval for a new RSV vaccine. This innovative vaccine, a recombinant formulation, is designed to actively immunize individuals aged 60 years and older, effectively preventing lower respiratory tract diseases caused by RSV [28]. Additionally, it can be administered to pregnant women to provide passive protection against RSV in infants from birth to 6 months of age, a benefit resulting from maternal immunization during pregnancy [29]. However, there is currently no vaccine available for RSV, and the development of an RSV vaccine for infants has proven to be notably challenging [30]. Research has demonstrated that natural RSV infection tends to elicit a weak immune response in children under 18 months of age [31]. Furthermore, infants under 4–6 months may struggle to develop a strong and lasting immune responses after vaccination [30]. Recently, a novel monoclonal antibody against RSV was developed: it was Nirsevimab, a single-dose of long-acting monoclonal antibodies that appear to be strongly effective in preventing RVS infection during the first winter season of life [32]. The phase 3 trial showed that Nirsevimab protection was maintained through 150 days after administration.

In Italy, since 2015, a preventive measure through monoclonal antibodies (palivizumab) has been in place to protect children at high risk of serious lower respiratory tract diseases caused by RSV, who require hospitalization. This preventive measure and its utilization are limited to a small subset of infants [32,33]: infants with a gestational age of 35 weeks or less who are less than 6 months old at the onset of the seasonal RSV epidemic; children younger than 2 years who have received treatment for bronchopulmonary dysplasia within the last 6 months; and children under 2 years of age diagnosed with hemodynamically significant congenital heart disease [34].

Numerous studies have provided evidence of significant reductions in RSV hospitalization rates among this highly vulnerable population, who are at elevated risk for RSV infection [33]. Palivizumab has half-life ranges from 19 to 27 days, necessitating monthly injections to sustain protection throughout the RSV season [35]. It is, therefore, clear that constant adherence to the monthly dosing schedule is crucial to maintain defense against RSV.

This analysis allowed us to assess the impact of hospitalization and factors associated with prolonged length of stay. In particular, our analysis showed that patients admitted to an intensive care setting experience a longer hospitalization stay, probably as this regimen is usually established for patients with an immediate life-threatening condition and consequently, a slower full recovery. Furthermore, evidences showed that prematurely born children and those with congenital heart disease and other genetic disorders have a higher risk of developing a severe RSV infection [36,37]. Our data have highlighted a strong association between these critical risk factors and increased length of hospitalization. Children with the above-mentioned clinical conditions are known to be at higher risk of being admitted to intensive care units [38,39]. Moreover, in our study, being older than 2 years was significantly associated with longer stay. This finding is partly in contrast with the existing evidence, as it is known that children by age 2 have an increased risk of developing a severe respiratory infection [40,41]. Further investigation will be necessary to explain the reason for this apparently discrepant result. However, the discrepancy between our findings and the existing literature can be attributed to our reliance solely on HDR admissions data, without considering emergency room admissions or urgent care flows [41]. It is conceivable that a share of patients did not generate discharge records because they were promptly transferred to specialized centers of excellence located in close proximity to the facility of Pescara. These centers, such as the IRCCS Bambino Gesù Hospital in Rome and the Salesi Hospital in Ancona, are renowned for their expertise in early childhood care and frequently absorb the great part of critically ill patients from near regions.

The results of this study must be read in light of some limitations. Firstly, results reflect the context of the single Local Health Authority of Pescara, a small province in southern Italy, which, therefore, does not exactly reflect the situation for the general population. Furthermore, as described above, the presence of two important centers of excellence for the management of pediatric patients, geographically not far from the Pescara hospital, could mean that the most critical patients were managed directly in these other structures. Secondly, the HDRs used were not originally compiled for epidemiological purposes but rather for admission-related remuneration. As a result, reported comorbidities in each record may have been subject to both overestimation and underestimation, or they could have been miscoded.

Thirdly, HDRs lack in crucial patient clinical data, such as drug therapies, blood parameters, and the clinical severity of each illness. The absence of this information could have potentially restricted the depth of our analysis [17]. In addition, socioeconomic status could not be evaluated, but it is a well-known predictor of bronchiolitis hospitalization. Finally, the use of HDRs focused only on hospitalized patients, lacking information on RSV cases managed out of the hospital setting. This situation could have led to an underestimation of the burden of disease.

Our study boasts several notable strengths. Firstly, it stands as the first investigation conducted within this region on bronchiolitis incidence. Furthermore, we extended our analysis over an extensive period spanning from 2015 to 2022. This prolonged timeframe afforded us a comprehensive and representative understanding of the disease. Lastly, the Pescara Local Health Authority remains a paramount healthcare service within the region, uniquely capable of providing care to patients from their first 30 days of life and beyond. This comprehensive approach encompasses both neonatal and pediatric care, encompassing sub-intensive and intensive care settings, including the Neonatal Intensive Care Unit and the Pediatric Intensive Care Unit. In addition, our study reported data on RSV during the pandemic year and during year 2022.

## 5. Conclusions

Bronchiolitis represents an important cause of hospitalization among patients aged less than 6 years. The incidence was particularly increased during the winter seasons in years 2021 and 2022.

## Figures and Tables

**Figure 1 diseases-12-00017-f001:**
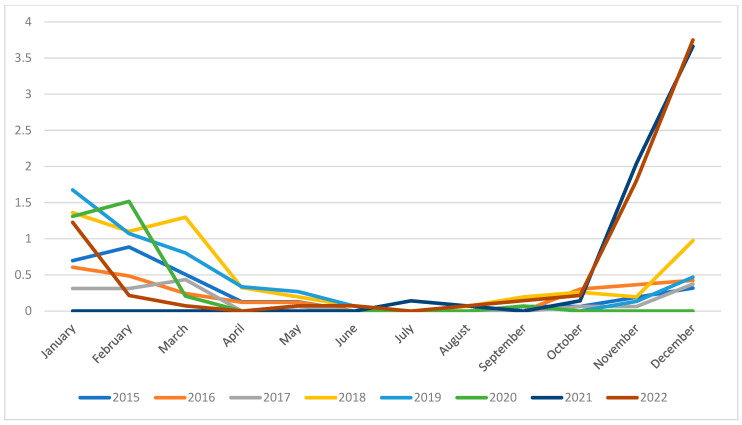
Admission rate (/1000 inhabitants) trend for bronchiolitis-related hospitalizations in Pescara province.

**Table 1 diseases-12-00017-t001:** Study population characteristics.

	Overall		Bronchiolitis	
	31,837		520	
Gender				
M	17,525	55.05%	298	57.31%
F	13,792	43.32%	222	42.69%
Age				
0	22,336	70.16%	455	87.50%
1	2901	9.11%	40	7.69%
2	2029	6.37%	15	2.88%
3	1735	5.45%	4	0.77%
4	1558	4.89%	3	0.58%
5	1278	4.01%	3	0.58%
Intensive Care				
Yes	1675	5.26%	459	88.27%
No	30,162	94.74%	61	11.73%
Risk Factors (premature, cardiac or congenital dysfunction)				
Yes	2326	7.31%	22	4.23%
No	29,511	92.69%	498	95.77%
RSV				
Yes	331	1.04%	189	36.35%
No	31,506	98.96%	331	63.65%
Death				
Yes	91	0.29%	0	0.00%
No	31,746	99.71%	520	100.00%

**Table 2 diseases-12-00017-t002:** Length of stay distribution comparisons between overall admissions and bronchiolitis-related admissions.

Length of Stay (Distribution by Days)		
Percentile of Children	All Admissions	Bronchiolitis Admissions
1%	0	0
5%	0	0
10%	1	0
25%	2	2
50%	3	4
75%	5	9
90%	10	20
95%	20	35
99%	95	216

**Table 3 diseases-12-00017-t003:** Negative binomial regression for bronchiolitis-related admissions.

	95% Conf. Interval		
	IRR *	Min	Max
Sex	1.00	0.90	1.12
Age > 2	5.48	4.49	6.68
Intensive Care	1.69	1.42	2.01
Risk Factors (Prematurity, cardiac or congenital dysfunction)	1.22	0.93	1.60
RSV	0.99	0.88	1.12

* adjusted for sex, age > 2, risk factors and RSV isolation.

## Data Availability

Data were not available due to policy restriction of Abruzzo Region.

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
