# Peer review of "Epidemiology of Bronchiolitis and Respiratory Syncytial Virus and Analysis of Length of Stay from 2015 to 2022: Retrospective Observational Study of Hospital Discharge Records from an Italian Southern Province before and during the COVID-19 Pandemic"

_diseases, 2024, doi:10.3390/diseases12010017_

Round 1

Reviewer 1 Report

Comments and Suggestions for Authors

Cedrone et al. present a relatively simple examination of the history of hospitalizations of young children for bronchiolitis in a southern Italian Province (Pescara) from 2015 to 2022. Although bronchiolitis is most often caused by RSV, is RSV the only virus that causes bronchiolitis?

Bronchiolitis appeared mostly from December to March. The authors state that the RSV season depends on the geographic location and the climate (l.78), as has been the theory for some time. But they found it each year except in the 2020-2021 season (COVID’s first year). It rebounded in the 2021-2022 and 2022- seasons to much higher levels. This pattern has been repeated in many regions of the world during this time.

The Results section is composed of three short paragraphs, 4 tables and one figure, described in 6 sentences. The last sentence notes the surge in bronchiolitis cases in 2021 and 2022, but not the absence of bronchiolitis in the 2020-2021 winter which would have left all children unprotected for the next year’s surge in cases.

In the Discussion, the authors mention the interruption of RSV circulation for that one year, and the surge of RSV cases in the two subsequent years. However, they do not describe what this finding tells us about the importance of geographic location and/or climate. They should. Daycare sites and schools were closed during that winter. Was the geographic location or the climate different? Close contact of children, not the climate, was clearly responsible for this gap year in the bronchiolitis/RSV.

The authors describe palivizumab and how it is used to protect some infants with underlying problems, but do not state if palivizumab was used to treat any of the infants in this study.

The statement on the last page: The data availability statement at the end of the manuscript: “Data were not available…” If data was not available, where did their numbers come from? Is this why bronchiolitis was used instead of laboratory detection of RSV infection?

The manuscript needs to be updated to include the now approved maternal vaccines and new monoclonal antibodies that will soon be used. They were not available during the study period, but they will likely be used in the future.

Comments on the Quality of English Language

Minor suggestions noted on the manuscript.

Author Response

  • Cedrone et al. present a relatively simple examination of the history of hospitalizations of young children for bronchiolitis in a southern Italian Province (Pescara) from 2015 to 2022. Although bronchiolitis is most often caused by RSV, is RSV the only virus that causes bronchiolitis?

Reply: RSV is not the only cause of bronchiolitis. This paper was based on administrative data from HDR. Frequently the exact pathogen was not reported in the discharge record.

  • The Results section is composed of three short paragraphs, 4 tables and one figure, described in 6 sentences. The last sentence notes the surge in bronchiolitis cases in 2021 and 2022, but not the absence of bronchiolitis in the 2020-2021 winter which would have left all children unprotected for the next year’s surge in cases.

Reply: We thank you the referee for the comment. We improved this section adding this important point. This point was also discussed in discussion section.

  • In the Discussion, the authors mention the interruption of RSV circulation for that one year, and the surge of RSV cases in the two subsequent years. However, they do not describe what this finding tells us about the importance of geographic location and/or climate. They should. Daycare sites and schools were closed during that winter. Was the geographic location or the climate different? Close contact of children, not the climate, was clearly responsible for this gap year in the bronchiolitis/RSV.

Reply: We thank you the referee for the comment. In Italy during year 2021 schools and living activities were re-opened and this point helped the recirculating of respiratory viruses. In addition, the described surge occurred during winter season,  confirming the impact of climate on bronchiolitis spread.

  • The authors describe palivizumab and how it is used to protect some infants with underlying problems, but do not state if palivizumab was used to treat any of the infants in this study.

Reply: We thank you the referee for the comment. As stated in the limitation section, HDR lack in some information as drug therapy performed during the admission.

  • The statement on the last page: The data availability statement at the end of the manuscript: “Data were not available…” If data was not available, where did their numbers come from? Is this why bronchiolitis was used instead of laboratory detection of RSV infection?

Reply: We corrected the sentence. The data cannot be shared due to privacy restriction.

  • The manuscript needs to be updated to include the now approved maternal vaccines and new monoclonal antibodies that will soon be used. They were not available during the study period, but they will likely be used in the future.

Reply: We thank you the referee for the comment. We improved the discussion section.

Reviewer 2 Report

Comments and Suggestions for Authors

This is a clearly presented retrospective assessment of hospitalizations for bronchiolitis including RSV bronchiolitis from a province in Southern Italy. The findings are as expected, with the surprise (also noted by others) of a reduction of this illness in the winter of 2020-2021.  You explain this well, with the hypothesis (which must be true!) that Covid 19 precautions reduced spread of other respiratory pathogens. The paper is valuable as an addition to our understanding of trends in bronchiolitis and RSV.  

Also well done is your explanation of your limitations: for example that some of the children with severe illness could have been admitted to care centers outside of your province and not included in your data.  But that limitation would have been consistent across the seven years you examine. 

This reviewer does not know whether the journal can have lengthy papers, but if they need a shorter paper, you can decrease your explanations of bronchiolitis and RSV in the introduction.  While you present this well, and in an edifying style, this information is widely available. Most important in your introduction are paragraphs 2, 3, 5 and 6 which pertain to your epidemiologic objectives.

You provide information on underlying illness among hospitalized children, and rates by population.  Missing from your report is information on socioeconomic status of the hospitalized children vs. the background population.    Perhaps that was not available in the records you had access to.

Specific comments:

Abstract

-It would be helpful to report the population of Pescara in the abstract. You do have that in the body of the manuscript.

-For the higher incidence rates in 2021 and 2022, 4.005 and 3.956, you could use one decimal place, 4.0 and 4.0, and one decimal place for the CIs.

Introduction 

-Para 2: is the 2019 data worldwide?

-Para 4: Information on prevention was not collected in your study and is not needed in the introduction, except perhaps as a brief comment.

Results:

-First para, typo: The table says 298 males not 222.

-Risk factors: Most had no underlying condition. Were there possibly socioeconomic stressors for the children without underlying conditions?

-table 2: this is difficult to read.  The left column (%) needs a header, something like "% of children " and the middle and 3rd columns should be shorter, with a hyphen between the lower and upper in the range of days, and also this needs a header, e.g. Range in number of days of hospital stay.

-Table 3: Do you mean age <2 not age >2?

-Table 4: this is also difficult to read.  It would be helpful to say "admission rates per _[what]____"? Population of children? percent of all hospitalizations in children?  The graph is much more informative. Perhaps Table 4 could be in an appendix.

Discussion:
-Para 1: again you are presenting "rates" but without explaining rate of what... e.g. Incidence of RSV infection was 26.2% : Out of what population. 

-Paragraph 5: Well said, re interruption of RSV transmission.

-Para 6-8: interesting and well presented but like the introduction, this information is available elsewhere and not derived from your data.  You could say what you need to more briefly. 

Since one of the most interesting aspects of your study is that it spans the worst of the Covid pandemic, you might add that to the title, e.g. epi trends in bronchiolitis and RSV before and during the covid pandemic. 

Comments on the Quality of English Language

The English language is mostly excellent, could use minor editing. For example, in introduction, line 70: "About ALRI RSV associated infection..."
 is an unusual way of introducing the topic.  Line 138: "the resident on January 1st was used" might better read "the population as of January 1..."

Author Response

  • It would be helpful to report the population of Pescara in the abstract. You do have that in the body of the manuscript.

Reply: Item addressed

  • For the higher incidence rates in 2021 and 2022, 4.005 and 3.956, you could use one decimal place, 4.0 and 4.0, and one decimal place for the CIs.

Reply: Item addressed

  • Introduction Para 2: is the 2019 data worldwide?

Reply: Yes it is. It is added in text.

  • Para 4: Information on prevention was not collected in your study and is not needed in the introduction, except perhaps as a brief comment.

Reply: We appreciate the reviewer comment but we think that information on the lack of preventive measure can be helpful in the explanation of the burden of disease.

  • First para, typo: The table says 298 males not 222

Reply: Item corrected;

  • Risk factors: Most had no underlying condition. Were there possibly socioeconomic stressors for the children without underlying conditions?

Reply: We thank the reviewer for the important point raised. As stated in the limitations section, the study analyzed HDR that lacks in several information such as drug therapy and laboratory test results. Also socio-economic indices cannot be derived from HDR, so we were not able to evaluate the impact of these conditions on hospitalizations. However, we add this point among limitations.

  • table 2: this is difficult to read.  The left column (%) needs a header, something like "% of children " and the middle and 3rd columns should be shorter, with a hyphen between the lower and upper in the range of days, and also this needs a header, e.g. Range in number of days of hospital stay.

Reply: Item corrected;

  • Table 3: Do you mean age <2 not age >2?

Reply: We are sorry for the mystake. We mean <2. Item corrected.

  • Table 4: this is also difficult to read.  It would be helpful to say "admission rates per _[what]____"? Population of children? percent of all hospitalizations in children?  The graph is much more informative. Perhaps Table 4 could be in an appendix.

Reply: Item corrected. The table was also shifted in appendix

  • Para 1: again you are presenting "rates" but without explaining rate of what... e.g. Incidence of RSV infection was 26.2% : Out of what population. 

Reply: It was the incidence rate of RSV hospitalizations in the cohort reported in the study at reference 18.

  • Paragraph 5: Well said, re interruption of RSV transmission.

Reply: Item corrected

  • Para 6-8: interesting and well presented but like the introduction, this information is available elsewhere and not derived from your data.  You could say what you need to more briefly. 

Reply: We appreciate the reviewer comment but, as also previously stated, we think that information on the lack of preventive measure can be helpful in the explanation of the burden of disease. Also, new frontiers in the RSV management can be useful to discuss in order to evaluate how the epidemiological data reported in this paper can change during next years

  • Since one of the most interesting aspects of your study is that it spans the worst of the Covid pandemic, you might add that to the title, e.g. epi trends in bronchiolitis and RSV before and during the covid pandemic. 

Reply: Title corrected

  • The English language is mostly excellent, could use minor editing. For example, in introduction, line 70: "About ALRI RSV associated infection..."

Reply: ALRI was previously defined in paragraph 1.

  • 138: "the resident on January 1st was used" might better read "the population as of January 1..."

Reply: Item ,corrected